# Propofol Affects Cortico-Hippocampal Interactions via β3 Subunit-Containing GABA_A_ Receptors

**DOI:** 10.3390/ijms21165844

**Published:** 2020-08-14

**Authors:** Matthias Kreuzer, Sergejus Butovas, Paul S García, Gerhard Schneider, Cornelius Schwarz, Uwe Rudolph, Bernd Antkowiak, Berthold Drexler

**Affiliations:** 1Department of Anesthesiology and Intensive Care, Klinikum rechts der Isar, Technical University of Munich, School of Medicine, Munich, Ismaninger Str. 22, 81675 München, Germany; m.kreuzer@tum.de (M.K.); gerhard.schneider@tum.de (G.S.); 2Werner Reichardt Centre for Integrative Neuroscience, Eberhard-Karls-University, Otfried-Müller-Str. 25, 72076 Tübingen, Germany; sergejus.butovas@googlemail.com (S.B.); cornelius.schwarz@uni-tuebingen.de (C.S.); 3Department of Anesthesiology, Neuroanesthesia Division, Columbia University Medical Center, New York Presbyterian Hospital, 622 West 168th Street, New York City, NY 10032, USA; pg2618@cumc.columbia.edu; 4Department of Comparative Biosciences, College of Veterinary Medicine, University of Illinois at Urbana-Champaign, 2001 South Lincoln Avenue, Urbana, IL 61802-6178, USA; urudolph@illinois.edu; 5Carl R. Woese Institute for Genomic Biology, University of Illiniois at Urbana-Champaign, Urbana, IL 61801, USA; 6Department of Anaesthesiology, Experimental Anaesthesiology Section, Eberhard-Karls-University, Waldhörnlestrasse 22, 72072 Tübingen, Germany; bernd.antkowiak@uni-tuebingen.de

**Keywords:** propofol, GABA_A_ receptor, cortex, hippocampus, local field potential, mutual information, phase locking, synchrony

## Abstract

Background: General anesthetics depress neuronal activity. The depression and uncoupling of cortico-hippocampal activity may contribute to anesthetic-induced amnesia. However, the molecular targets involved in this process are not fully characterized. GABA_A_ receptors, especially the type with β3 subunits, represent a main molecular target of propofol. We therefore hypothesized that GABA_A_ receptors with β3 subunits mediate the propofol-induced disturbance of cortico-hippocampal interactions. Methods: We used local field potential (LFP) recordings from chronically implanted cortical and hippocampal electrodes in wild-type and β3(N265M) knock-in mice. In the β3(N265M) mice, the action of propofol via β3subunit containing GABA_A_ receptors is strongly attenuated. The analytical approach contained spectral power, phase locking, and mutual information analyses in the 2–16 Hz range to investigate propofol-induced effects on cortico-hippocampal interactions. Results: Propofol caused a significant increase in spectral power between 14 and 16 Hz in the cortex and hippocampus of wild-type mice. This increase was absent in the β3(N265M) mutant. Propofol strongly decreased phase locking of 6–12 Hz oscillations in wild-type mice. This decrease was attenuated in the β3(N265M) mutant. Finally, propofol reduced the mutual information between 6–16 Hz in wild-type mice, but only between 6 and 8 Hz in the β3(N265M) mutant. Conclusions: GABA_A_ receptors containing β3 subunits contribute to frequency-specific perturbation of cortico-hippocampal interactions. This likely explains some of the amnestic actions of propofol.

## 1. Introduction

Commonly used general anesthetics such as propofol influence information processing in the brain. Hypnotic doses of this anesthetic, for instance, impair the transfer of information between the frontal and parietal cortical regions [1,2,3] as well as between the thalamus and the cortex [4,5]. While impairment of thalamocortical activity may relate to the hypnotic component of anesthesia, propofol-induced changes in cortico-hippocampal interactions may contribute to the amnestic component of this drug. Memory-related processes are strongly linked to interactions between these areas [6]. Both regions interact with each other via neural oscillating activity in the mid-frequency (4–12 Hz) range [7]. Volatile anesthetics cause a slowing of the peak oscillation frequency in the hippocampus. This may represent one mechanism of anesthetic-induced amnesia [8,9]. It is difficult to determine which specific molecular pathways are involved in this process, because volatile anesthetics cause many changes via multiple molecular targets in the brain, e.g., potassium channels, HCN channels, acetylcholine receptors, glutamate receptors and GABA_A_ receptors. [10,11,12,13]. Compared to volatile anesthetics, propofol is much more specific for GABA_A_ receptors [11,14]. The anesthetic actions of propofol are mediated almost exclusively via βsubunit containing GABA_A_ receptors [15,16]. Using genetically engineered mice, it was shown that β2subunit-containing GABA_A_ receptors, a frequently expressed subtype of GABA_A_ receptors [17] are involved in the sedative properties of intravenous anesthetics [18,19]. On the other hand, β3 subunit-containing GABA_A_ receptors, being only a minor subtype of GABA_A_ receptors [17], largely contribute to the hypnotic and immobilizing effects of propofol [20]. Furthermore, the electrophysiological effects of propofol in the cerebral cortex critically depend on the presence of β3 subunit-containing GABA_A_ receptors [21]. Therefore, we used wild-type and β3(N265M) knock-in mutant mice to elucidate the impact of β3 subunit-containing GABA_A_ receptors on the effects of propofol on cortico-hippocampal interactions. The β3(N265M) mice are largely resistant to the actions of propofol at this receptor-subtype [22]. We hypothesized that the GABAergic anesthetic propofol would reduce connectivity between the cortex and hippocampus and that this reduction in connectivity is at least in part mediated by β3 subunit-containing GABA_A_ receptors. We tested this hypothesis by (i) analyzing the local field potential (LFP) spectral power in (prefrontal) cortex and hippocampus, (ii) the interaction between these brain areas using phase locking and (iii) the shared information content using the mutual information.

## 2. Results

We did not observe significant differences in body weight of the animals from both groups. (WT: 33.4 ± 5.0 g vs. β3(N265M): 36.4 ± 5.9 g). Animals were recorded under awake control conditions for ten minutes, injected i.v. with propofol, and immediately monitored again until they showed recovery from anesthesia in their LFP recordings. Head-fixation impeded with simultaneous test of return of righting reflex (RoRR). Therefore, time to RoRR after propofol injection was measured in separate experiments. We determined RoRR after approximately 10 min in wild-type and 2 min in β3(N265M)-mutant mice.

### 2.1. Effects of Propofol on the Power Spectrum of Cortical and Hippocampal Oscillations in Wild-Type and β3(N265M) Knock-in Mice

Propofol did not change spectral power in the 4–12 Hz range at cortical recording sites in wild-type mice, yet there was a significant increase in power in frequencies above 12 Hz. This increase in higher-frequency power was also observed in data recorded from hippocampal sites. The most striking difference between the hippocampus and cortex, however, was a strong activation of hippocampal but not of cortical oscillations in the 4–6 Hz range. For the β3(N265M) mice, the results are different. Propofol did not have a significant effect on the spectral band power in cortex. The activation of 4–6 Hz oscillatory activity we observed in the wild-type mice in the hippocampus was also absent in the β3(N265M) animals. However, in β3(N265M) mice, we observed a small but significant increase in 8–10 Hz band power. The detailed effects of propofol on spectral band power are presented in Figure 1.

### 2.2. Actions of Propofol on Interactions between Cortex and Hippocampus in Wild-Type and β3(N265M) Knock-in Mice

In the next step, we focused on changes in interactions between the cortex and hippocampus using the phase locking value (PLV). In wild-type mice, we found a strong de-synchronization of phase in the 6 to 12 Hz range, indicating that propofol impairs the oscillatory connectivity between the cortex and hippocampus. The β3(N265M) knock-in mice did not show that strong a decrease in PLV. In these animals, PLV showed a significant and strong decrease only in a smaller frequency window from 6–8 Hz. Figure 2 displays the PLV for the wild-type and β3(N265M) mice for all frequency ranges. This demonstrates that GABA_A_ receptors containing the β3 subunit might be involved in communicating neuronal information between the cortex and hippocampus in the 8–14 Hz range while receiving propofol sedation/anesthesia.

### 2.3. Effects of Propofol on Shared Information Content between Cortex and Hippocampus in Wild-Type and β3(N265M) Knock-in Mice

Finally, the use of mutual information (MI) served the purpose to estimate propofol-induced changes of shared information between cortex and hippocampus. In wild-type mice, propofol caused a strong reduction in shared information in frequencies from 6–16 Hz. In contrast, MI in β3(N265M) knock-in mice did not show a decrease to that degree, and the effect was limited to a smaller frequency range from 6 to 8 Hz. This is in line with the finding for PLV, where the effects in the β3(N265M) knock-ins were attenuated, i.e., limited to a smaller frequency range as well. Figure 3 shows the effect of propofol on MI in the single frequency ranges for wild-type and β3(N265M) knock-in mice.

The Hedges’ g values and their CI for all experiments (power spectral density (PSD), PLV, and MI) in the single frequency intervals are listed in Table 1.

## 3. Discussion

We present results regarding different effects of propofol on cortico-hippocampal interactions in two animal models, i.e., in wild type and β3(N265M) knock-in mice. The animals showed an attenuated response to propofol. Investigations of anesthetic-induced modulation of interactions between brain regions have been in the focus of research for some decades. In the mid-1990s, MacIver and colleagues demonstrated that the intravenous anesthetic thiopental affects cortico-hippocampal synchronicity in a dose-dependent manner [23]. More recently, Perouansky and co-workers showed that volatile anesthetics slow hippocampal oscillations in the 4–12 Hz range, thereby inducing amnesia [8]. This knowledge is extended by the present study by showing that the GABAergic anesthetic propofol also affects hippocampal oscillatory activity in this frequency range, weakens cortico-hippocampal interactions and lowers the shared information content between these two brain regions. Based on the observed differences between wild-type and β3(N265M) mutant mice, we conclude that this process critically depends on GABA_A_ receptors containing β3 subunits.

Our experiments served the purpose to investigate propofol-induced effects on interactions between cortex and hippocampus in the range from 4 to 12 Hz that are strongly related to processes that are necessary for memory consolidation [24]. The cortex and the hippocampus play an important role in memory formation. The observed effects of propofol on oscillatory activity in these regions are likely to be associated with the amnestic component of anesthesia. Rodent studies have also demonstrated that parts of the frontal cortex and hippocampus are among the cerebral regions that are most sensitive to anesthetics [11]. The actions of propofol in the CNS are almost exclusively mediated via GABA_A_ receptors containing β subunits [15,25]. There are three different isoforms of β subunits (β1-β3), resulting in at least three different subtypes of GABA_A_ receptors as defined by their β subunits [26,27]. Currently, studies measuring behavioral endpoints of the actions of propofol on β1 subunit-containing GABA_A_ receptors are limited. However, roughly a decade ago, two landmark studies by Reynolds et al. [18] and Jurd et al. [20] provided evidence that sedation by intravenous anesthetics is primarily mediated by β2 subunit-containing GABA_A_ receptors, while hypnosis and immobility involves β3 subunit-containing GABA_A_ receptors. Furthermore, findings using cortical slice cultures from β3(N265M) knock-in mice demonstrate that the typical electrophysiological actions of propofol—the prolongation of inhibitory postsynaptic currents—critically depend on the presence of the β3 subunit-containing GABA_A_ receptor subtype [21]. With this in mind, we evaluated the actions of propofol on cortico-hippocampal interactions in wild-type and β3(N265M) knock-in mice to elucidate the impact of β3 subunit-containing GABA_A_ receptors.

As mentioned above, the cortex is strongly associated with working, i.e., short-term, memory tasks, but also requires interactions with other brain areas like thalamus and hippocampus [28,29]. In order to consolidate memory, the hippocampus is required. Siapas et al. described that activity in the 4–10 Hz frequency range seems important to gate information flow across cortico-hippocampal circuits. The subsequent plastic changes are believed to underlie the storage of information across these networks [30].

We structured our experiments in three steps. The first investigation was to evaluate propofol-induced changes in spectral band-power in the cortex and hippocampus. The analysis of spectral band-power revealed that propofol does not affect oscillations in the mid-frequency range in the cortex. This result complies with findings from other studies showing that propofol, at least at low concentrations, does not affect the (prefrontal) cortex [31]. In our experiments in wild-type mice, propofol caused a strong activation in the 4–6 Hz range, relating to the theta range, in the hippocampus. This observation is in line with previous experimental findings. Perouansky et al. discovered a slowing of oscillations in the 4–12 Hz range in the rat’s hippocampus as an effect of subanesthetic doses of inhaled anesthetics. It was hypothesized that this slowing led to impaired learning. The authors concluded that modulating hippocampal θ-oscillations may present a key process of anesthetic-induced amnesia [8]. Pan et al. found a crucial frequency of around 6.5 Hz for hippocampal oscillations. If peak frequency drops below 6.5 Hz, learning impairments occur. This may be due to a slowing of these θ-frequencies or due to a reduction in high frequency θ-oscillations [32]. The activation of low θ-frequencies was absent in the β3(N265M) knock-in mice. This finding may indicate that β3 subunit-containing GABA_A_ receptors, which are heavily expressed in both the cortex and hippocampus [33,34,35,36] are necessary to mediate propofol-induced amnestic actions.

The use of PLV can help to determine effects on oscillation synchrony between cortex and hippocampus and the concept of MI may help to detect changes in the amount of information that is shared by these regions through oscillatory activity in the selected frequency range. The next step consisted of an analysis of cortico-hippocampal interactions using PLV to evaluate phase synchrony and hence the strength of connectivity between the two recording sites. This synchrony may be necessary for interactions between different cerebral structures [37]. Recently, a study using PLV showed that phase coupling in the 3–9 Hz range between the cortex and visual area after a visual stimulus is enhanced during active working memory processes [38]. Furthermore, synchrony between the cortex and hippocampus increases during spatial working memory tasks [39], underlining the importance of synchrony for mnemonic processes. Our analysis was targeted at cortico-hippocampal interactions, possibly reflecting memory consolidation processes rather than working memory activity. Propofol negatively affected cortico-hippocampal PLV in the 6–12 Hz range, i.e., synchrony and hence strength of connection decreased. Propofol also negatively influenced PLV in β3(N265M) knock-in mice, but to a lesser degree as compared to the wild-type, since only PLV in the 6–8 Hz range was significantly reduced.

MI analysis indicated significantly reduced shared information content between the cortex and hippocampus with propofol in the 6–16 Hz oscillation range in wild-type animals. This reduction may be explained by the observed drop in PLV at these very frequencies with propofol. Synchrony and hence possibly the strength of cortico-hippocampal interaction become reduced. This reduction consequently leads to reduced information that is shared between both regions. The shared information is also reduced by propofol in the β3(N265M) mice, but to a lesser degree and in a narrower frequency range as in the wild-types.

### Limitations

First of all, we are aware of our small sample size. Nevertheless, we found significant and strong effects that highlight the differences in LFP activity between the strains. The delivery of a single bolus dose of propofol via the tail vein leads to a peak and then decreasing concentrations of propofol in the brain. Hence, we did not investigate the effects at steady state conditions, which is very difficult to do in mice, but rather at selected, artifact-free and non-suppressed episodes that followed the (burst) suppression states after propofol delivery. Our findings highlight differences in LFP activity between the wild type and β3(N265M) mutant mice. LFP effects at distinct propofol concentrations need to be investigated in the future.

We also could not investigate the return of the righting reflex, because the head of the animals was fixated. We also did not perform behavioral experiments to investigate memory performance. This should also be subject to further investigations. Finally, we focused on studying a single point mutation in the β3 subunit of the GABA_A_ receptor. The β3 subunit was selected, as the β3-containing GABA_A_ receptors have been shown to be of major importance for the molecular and behavioral actions of propofol [20,22].

## 4. Materials and Methods

Every endeavor was made to minimize both the suffering and the number of mice sacrificed for the study. All experiments and procedures performed were approved by the local animal care committee (1 January 2007–31 December 2010, Eberhard-Karls-University, Tübingen, Germany) and in accordance with the German law on animal experimentation. Wild-type (*n* = 4) and GABA_A_ receptor β3(N265M) knock-in mice (*n* = 3) of the same genetic background (87.5% 129 × 1/SvJ, 12.5% 129/Sv) of both sexes were used [20]. The β3(N265M) point mutation renders β3 subunit-containing GABA_A_ receptors insensitive to propofol, but not towards its natural ligand GABA [20,22].

### 4.1. Surgical Procedures

The electrode implantation procedure was described in detail in previous publications [28,40,41,42,43]. During general anesthesia with isoflurane, a warming apparatus (Fine Science Tools, Heidelberg, Germany) kept body temperature close to 36 °C. The electrode arrays, consisting of four linear electrodes, were implanted after craniotomy using a stereotactic frame, two arrays in each animal on one hemisphere. The cortical array was placed 200 µm below the pia, the other array was placed 900 µm below the pia to target the hippocampus. Electrode arrays were fixed along with their connecting wires to the skull using dental cement (Flowline, Heraeus Kulzer, Hanau, Germany). Finally, the skin incision was sutured and attached to the implant. Post-operatively, the animals were kept warm and analgesia was achieved using carprofen.

### 4.2. Electrophysiology

Custom-built microelectrode arrays were used. A 1 × 4 array of polyimide tubing (HV Technologies, Trenton, GA, USA) served as a frame for electrodes made of platinum and tungsten that was glass coated. Electrodes (Thomas Recording, Giessen, Germany) had a shank diameter of 80 µm, a metal core diameter of 23 µm, and a free tip length of 10 µm and impedance was higher than 1 MΩ. The inter-tip distance in the assembled arrays was around 300 µm. Teflon-insulated silver wires (Science Products, Hofheim, Germany) were soldered to the electrodes on one end and connected to a micro-plug (Bürklin, Munich, Germany). Local field potentials were recorded using a multichannel extracellular amplifier (MultiChannelSystems, Reutlingen, Germany; gain 5000, digital sampling rate 20 kHz) in a head-restraint apparatus.

### 4.3. Recording and Pre-Processing of Local Field Potentials

Adult wild-type and β3(N265M) mice were housed individually for at least one week in a room maintained on a 12-hr light/dark cycle with free access to food and water before electrophysiological recordings. Propofol (10 mg/kg, B.Braun, Melsungen, Germany) was injected intravenously through the tail vein. Electrophysiological recording was started after placing the animal in a head-restraint apparatus. For each experiment, neuronal activity was monitored for ten minutes immediately before propofol treatment (baseline condition) and after propofol administration. The wild-type mice entered a state of isoelectric suppression and required around 50 min to return to baseline LFP activity. A similar state was reached by the mutant mice, already at around 20 min. In the effective period, LFP burst suppression was observed, without ever developing isoelectric LFP.

After the termination of the experiments, the mice were deeply anesthetized and an intracardial perfusion of 0.1 M phosphate puffer prior to 4% paraformaldehyde in phosphate buffer was delivered. The recording sites were marked by electrolytic lesions of the brain tissue, which were identified in the Nissl-stained coronal sections [28].

The initial step of signal preprocessing contained a 100 Hz low pass filtering with a MATLAB (Mathworks, Natick, MA, USA)-based Butterworth filtering routine and a down sampling from *f_s_* = 20 kHz to *f_ds_* = 250 Hz. We present all used LFP episodes in Figure A1. For analysis in the distinct frequency ranges, the recordings were additionally band pass filtered with the MATLAB routine. We used the *filtfilt* function, allowing filtering without phase shift. Then, we extracted a simultaneous artifact-free 5 s LFP episode from each of the eight channels from all experiments for analysis. In our analyses, we focused on the 2–16 Hz frequency range of the LFP. In order to evaluate effects in the single oscillation frequencies, we investigated oscillatory activity in non-overlapping 2 Hz frequency bands.

### 4.4. Spectral Power

We used MATLAB to calculate the relative band power for each frequency band represented in the histogram. First, the power spectral density of the 5 s LFP episode over the entire frequency range and for each channel was estimated using the fast Fourier transform. The absolute power spectral density vector was divided through its sum to obtain the relative power. The sum of relative power in the desired frequency range represents the relative band power. Relative band power was calculated for both regions, cortex and hippocampus, to investigate propofol induced effects.

### 4.5. Phase Locking

We used the phase locking value (PLV) to investigate interactions between cortex and hippocampus. Lachaux et al. introduced PLV to detect synchronic behavior in a defined frequency range between two channels [37]. PLV calculation is based on the analytic phase *φ(t)* relationship between both channels. *φ(t)* describes the momentary position of the oscillation for each data point. In contrast to Lachaux, we used the Hilbert transform [44] instead of the wavelet approach to calculate *φ(t)*. PLV is based on the difference in analytic phase between two channels, here the difference in the hippocampal and prefrontal cortical analytic phase, *φ_hippocampus_(t)* and *φ_cortex_(t)*. The details are described in the next section. Figure A2 shows exemplary signals that are either completely phase locked or completely independent.

### 4.6. Description of Hilbert Transform and PLV

The analytic signal *X(t)* of a recorded and filtered signal *x(t)*, here the envelope of the LFP in the respective frequency range, can be derived with the Hilbert transform [44]. In order to correctly apply this method, the LFP has to be filtered to a narrow frequency range [45], here the described frequency bins of 2 Hz width.
$$X(t)=x_{RE}(t)+x_{IM}(t)$$
is a series of complex numbers with *x_RE_(t)* as the real part and *x_IM_(t)* as the imaginary part.

The real part *x_RE_(t)* complies with the original signal *x(t)* and *x_IM_(t)* complies with *x(t)* after a 90° phase shift. Based on *X(t)*, the analytic amplitude *A(t)* and the analytic phase *φ(t)* can be calculated.

*A(t)* represents the envelope of *x(t)* and is derived by $A(t)=\sqrt{x_{RE}{\left(t\right)}^{2}+x_{IM}{\left(t\right)}^{2}}$.

*φ(t),* also called the instantaneous phase, is calculated as $\phi (t)=\arctan(x_{IM}(t)/x_{RE}(t))$.

The difference Φ(t) is defined as $\Phi (t)=\phi_{hippocampus}(t)-\phi_{cortex}(t)$

PLV can be consequently calculated according to $PLV=\frac{1}{N}\sum \exp(i\Phi (t))$ where N represents the number of data points in the time series.

PLV ranges between 1 and 0 and high values indicate little variance in phase difference, i.e., high synchronization and possibly strong connectivity, whereas a PLV close to 0 indicates that the signals are not connected.

We calculated and averaged the PLV from the extracted 5 s LFP episodes over 5 non-overlapping 1 s (N = 250) segments. Figure A2 presents a graphical description of the PLV calculation.

### 4.7. Mutual Information

Mutual information (MI) evaluates shared information content between two recording sites [46] i.e., the dependence of two channels. The higher the MI, the higher the dependence between two channels. An MI of zero describes two independent channels. MI was calculated over the entire 5 s segments for all hippocampal-cortical channel combinations.

MI calculation is based on the evaluation of probability distributions—in our case, either distribution of LFP amplitudes of a sequence of LFP amplitudes in both selected channels.

Therefore, in a first step, the filtered 5 s LFP sequences were normalized, i.e., presented as z-scores according to $x_{n}\left[t\right]=\frac{x\left[t\right]-mean(x\left[t\right])}{std(x\left[t\right])}$. *x[t]* is a discrete time series, here the filtered and down sampled 5 s LFP episode. In order to generate the amplitude-based probability distributions and the LFP amplitude values of the normalized series *x_n_[t],* we used 10 equally spaced bins, as in Figure A3. The width of one bin was the maximum amplitude range, i.e., max(*x_n_[t]*)-min(*x_n_[t]*) divided by the number of bins. For each of the two channels, *x_n_[t]* and *y_n_[t]*, the probability distributions *p(x)* and *p(y)* are calculated based on their bin histograms as displayed in Figure A3. Furthermore, the joint probabilities *p(x,y)* are determined. The mutual information can be estimated by:
$MI=\sum_{x\in N}\sum_{y\in N}p(x,y)\log\left(\frac{p(x,y)}{p(x),p(y)}\right)$, where *N* is the number of bins [47].

### 4.8. Statistics

We used the Hedge’s g effect size measures for dependent and independent samples. Calculation was performed with the MATLAB based MES toolbox [48]. We also performed a 10,000-fold bootstrapping to generate 95% confidence intervals (CI)—95% CI excluding zero indicates significance at *p* < 0.05. As a rule of thumb, absolute Hedge’s g values > 0.8 indicate a strong effect and values above 0.5 indicate a medium effect [49]. Hence, we discuss the results from the Hedge’s g analyses according to this notation, i.e., we describe a strong effect if g > 0.8 and a medium effect if g > 0.5.

## 5. Conclusions

Our results suggest that the observed propofol-induced changes in interactions between cortex and hippocampus are largely mediated by β3 subunit-containing GABA_A_ receptors and that this GABA_A_ receptor subtype potentially participates in the process of inducing amnesia by propofol. The decrease in spectral band-power in the 4–6 Hz range was absent in β3(N265M) knock-in mice and the reduction in PLV and MI in our analyses of the θ-frequency range was not as pronounced as in the wild-types.

## Figures and Tables

**Figure 1 ijms-21-05844-f001:**
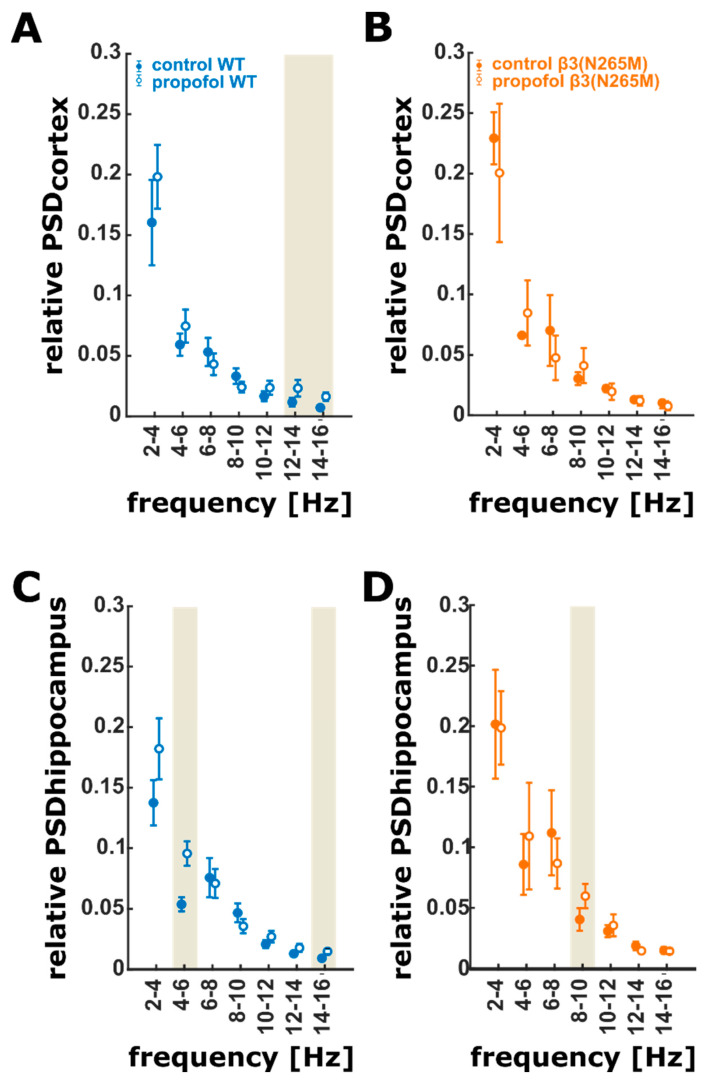
PSD error bar plots of the 2 Hz frequency ranges (mean ± sem). (**A**) LFP recorded from the cortex in the wild-type mice. Propofol has a significant and medium effect on the 12–14 Hz and a significant and strong effect on the 14–16 Hz (Hedges’ g: 1.01; CI: (0.38 to 1.96)) frequency range. In both frequency ranges, propofol causes an increase in relative power. (**B**) There is no significant effect of propofol on the PSD of LFP activity recorded from cortex in the β3(N265M) group. (**C**) In the hippocampal LFP recordings from wild-types, propofol significantly and strongly increased relative power in 4–6 Hz range and in the 14–16 Hz range (Hedges’ g: 0.90; CI: (0.26 to 1.78)). (**D**) Propofol caused a significant and strong increase in relative power in the 8–10 Hz range of LFP recorded from the hippocampus of the β3(N265M) mice. The ivory areas indicate significance, i.e., a 95% confidence interval of Hedge’s g exclusive 0; wild-type (*n* = 4), β3(N265M) knock-in mice (*n* = 3); PSD: power spectral density. LFP: local field potential, CI: 95% confidence interval.

**Figure 2 ijms-21-05844-f002:**
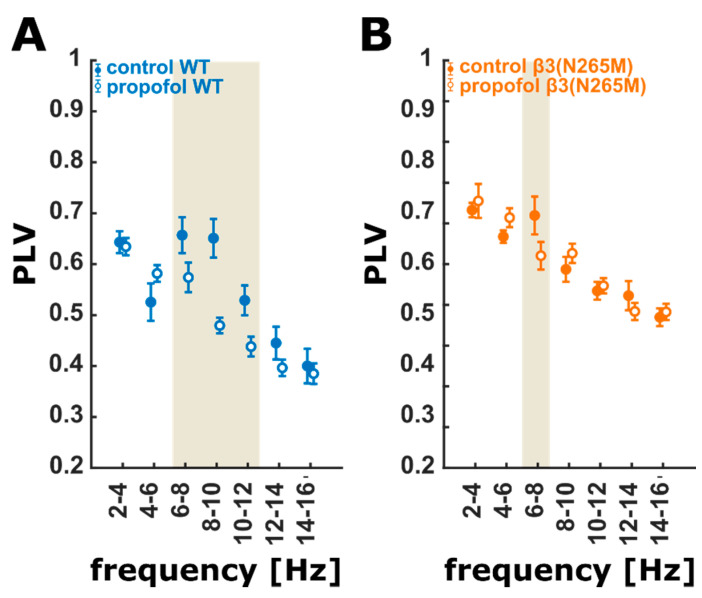
PLV for the wild-type (**A**) and β3(N265M) mutant recordings (**B**) for non-overlapping 2 Hz frequency bands. (**A**) Propofol strongly and significantly affects PLV in the 6–12 Hz range in the wild-type group. (**B**) Propofol only shows a significant (strong: g = 1.06; CI: (0.38, 2.85)) effect on PLV in the 6–8 Hz range in the β3(N265M) mutant group. The ivory areas indicate significance, i.e., a 95% confidence interval of Hedge’s g exclusive 0; wild-type (*n* = 4), β3(N265M) knock-in mice (*n* = 3); PLV: phase locking value; CI: 95% confidence interval.

**Figure 3 ijms-21-05844-f003:**
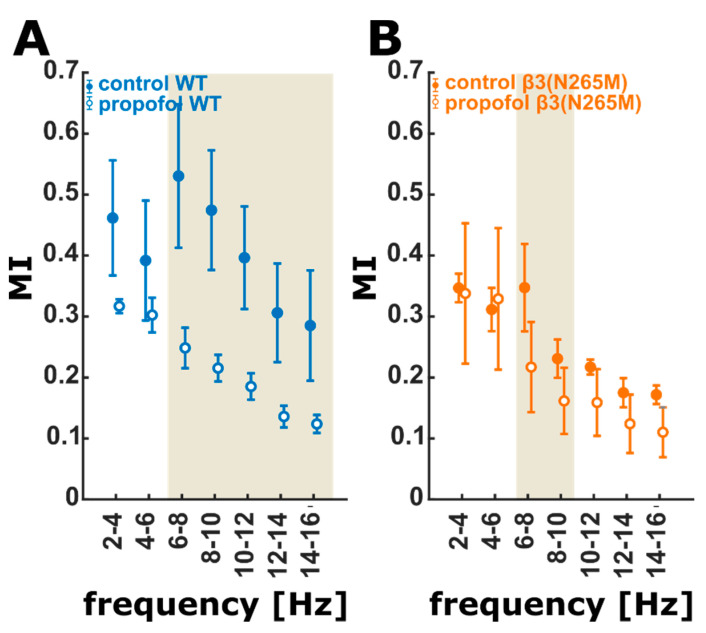
Propofol-induced change in mutual information MI (10 bins) in wild-type (**A**) and β3(N265M) mice (**B**) at control conditions and in the presence of propofol. (**A**) In the wild-type analysis, propofol significantly reduces MI in frequencies above 6 Hz. The observed effects are classified as strong effects by the Hedges’ g test. (**B**) In the β3(N265M) group propofol has a significant (medium: g < 0.8) decreasing effect on MI only in the frequency range of 6–10 Hz. The ivory areas indicate significance, i.e., a 95% confidence interval of Hedge’s g exclusive 0; wild-type (*n* = 4), β3(N265M) knock-in mice (*n* = 3); MI: mutual information.

**Table 1 ijms-21-05844-t001:** The table shows the Hedges’ g values and corresponding 95% CI intervals of the comparison of spectral power, PLV, and MI at control conditions and in the presence of propofol. Grey entries are not significant. A bold representation in black indicates a significant strong effect. A bold and italic representation in black indicates a significant medium effect. PLV: phase locking value; MI: mutual information; CI: 95% confidence interval.

Type	Frequency	Hedge’s g (95% CI)			
		*Spectral Power*		*PLV*	*MI*
		*Cortex*	*Hippocampus*		*10 Bins*
**Wild-type**	***4–6 Hz***	−0.40 (−1.25, 0.32)	**−1.55 (−2.68, −1.05)**	−0.60 (−2.31, 0.18)	0.39 (−1.13, 1.19)
	***6–8 Hz***	0.29 (−0.45, 1.04)	0.10 (−0.71, 0.65)	***0.78 (0.21, 1.60)***	**1.04 (0.39, 1.90)**
	***8–10 Hz***	0.50 (−0.29, 1.41)	0.48 (−0.53, 1.39)	**1.80 (1.20, 3.01)**	**1.16 (0.68, 2.06)**
	***10–12 Hz***	−0.42 (−1.07, 0.10)	−0.46 (−1.17, 0.22)	**1.11 (0.68, 1.88)**	**1.09 (0.80, 1.81)**
**β3(N265M)** **Mutant**	***4–6 Hz***	0.29 (−1.02, 1.98)	−0.28 (−0.70, 0.16)	−1.03 (−4.30, 0.33)	−0.09 (−2.21, 0.85)
	***6–8 Hz***	−0.42 (−2.93, 1.02)	0.38 (−0.16, 1.54)	**1.06 (0.38, 2.85)**	***0.78 (0.03, 4.69)***
	***8–10 Hz***	0.40 (−1.23, 1.31)	**−0.87 (−3.50, −0.69)**	−0.62 (−3.49, 0.25)	***0.68 (0.59, 1.84)***
	***10–12 Hz***	−0.43 (−6.27, 0.57)	−0.28 (−1.12, 0.95)	−0.27 (−2.66, 8.03)	0.64 (−0.18, 2.34)

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
