# Peer review of "Propofol Affects Cortico-Hippocampal Interactions via β3 Subunit-Containing GABAA Receptors"

_ijms, 2020, doi:10.3390/ijms21165844_

Round 1

Reviewer 1 Report

This interesting study helps to tease apart GABAA receptor subtype contributions to the amnesic effects produced by propofol in mice.  Propofol is shown to cause an increase in beta frequency power in wild-type mice, that is lacking in β3 knock outs, thus demonstrating the importance of these subunits in the amnesic effect.  Through care analyses of phase locking between cortex and hippocampus, as well as mutual information coupling, the Authors demonstrate a break-down in communication between these brain regions that are well known to participate in memory formation.  I have a few minor suggestions for improvement of the manuscript.

“GABA. receptors containing β3” should be “GABAA receptors containing β3”

“This may explain the amnestic actions of propofol at least in part.”  Could be “This likely explains some of the amnestic actions of propofol.”

“Volatile anesthetics cause a slowing of the peak oscillation frequency in the hippocampus.”  Please cite MacIver & Bland 2014 (e.g. fig 1) for showing isoflurane slows both cortical and hippocampal oscillation frequencies.

“brain areas like thalamus or hippocampus” should be “brain areas like thalamus and hippocampus”

“The next step Consisted of an analysis” should be “The next step consisted of an analysis”

“artifact-free and non-suppressed episodes”  Did propofol produce burst-suppression following your administration?  If so you should probably mention this.

“LFP effects at distinct propofol concentration to LFP need to be investigated in the future.” Should be “LFP effects at distinct propofol concentrations need to be investigated in the future.”

Figure axis and symbol labels need to use increased font sizes for clarity.

Author Response

Dear Norah Tang,

Thank you for sending us the reviewers’ comments on our manuscript ijms-891696 entitled "Propofol affects cortico-hippocampal interactions via β3
subunit-containing GABAA receptors".

We thank the reviewers for providing valuable advice.  Here we present a revised version of our manuscript, which addresses the weaknesses pointed out by the reviewers. All changes in the manuscript are highlighted by using the “Track Changes” function.

We would like to answer the reviewer’s comments as follows:

Reviewer #1:

This interesting study helps to tease apart GABAA receptor subtype contributions to the amnesic effects produced by propofol in mice.  Propofol is shown to cause an increase in beta frequency power in wild-type mice, that is lacking in β3 knock outs, thus demonstrating the importance of these subunits in the amnesic effect.  Through care analyses of phase locking between cortex and hippocampus, as well as mutual information coupling, the Authors demonstrate a break-down in communication between these brain regions that are well known to participate in memory formation.  I have a few minor suggestions for improvement of the manuscript.

“GABA. receptors containing β3” should be “GABAA receptors containing β3”

 Thanks for spotting. We changed it.

“This may explain the amnestic actions of propofol at least in part.”  Could be “This likely explains some of the amnestic actions of propofol.”

Thanks for this suggestion. We changed the sentence accordingly.

“Volatile anesthetics cause a slowing of the peak oscillation frequency in the hippocampus.”  Please cite MacIver & Bland 2014 (e.g. fig 1) for showing isoflurane slows both cortical and hippocampal oscillation frequencies.

We apologize for not having included this reference in the initial version. We now cite this reference.

“brain areas like thalamus or hippocampus” should be “brain areas like thalamus and hippocampus”

Changed.

“The next step Consisted of an analysis” should be “The next step consisted of an analysis”

Corrected.

“artifact-free and non-suppressed episodes”  Did propofol produce burst-suppression following your administration?  If so you should probably mention this.

 Thanks for this suggestion. We already stated this in the methods section 4.3 as “The wild-type mice entered a state of isoelectric suppression and required around 50 min to return to baseline LFP activity. A similar state was reached by the mutant mice, already at around 20 minutes. In the effective period, LFP burst suppression was observed, without ever developing isoelectric LFP”

To make it more clear, we added “that followed the (burst) suppression states after propofol delivery” at the end of the mentioned sentence in the Limitations section.

“LFP effects at distinct propofol concentration to LFP need to be investigated in the future.” Should be “LFP effects at distinct propofol concentrations need to be investigated in the future.”

Changed.

Figure axis and symbol labels need to use increased font sizes for clarity.

We increased the fonts.

Reviewer 2 Report

This study carried out by Kreuzer et al. investigated the potential molecular target of amnesia evoked by the general anesthetic propofol.  By recording local field potential (LFP) from the cortex and hippocampus in wild type and b3(N265M) knock-in mice, they observed that propofol differentially affects the power spectral density, phase locking value and mutual information in these two groups of mice.  The authors concluded that the frequency-specific perturbation of cortico-hippocampal interactions contributes, at least in part, to amnestic effect of propofol and that this effect is dependent on the GABAA receptor b3 subunit.  This is an interesting study.  The manuscript is well organized and presented. 

However, previous electrophysiology studies indicate that b2/b3 N265M mutation eliminates etomidate sensitivity and reduces propofol effects in a1b2/3g2 GABAA receptors.  Why was propofol instead of etomidate chosen in the current study?  It would be interesting to know if etomidate engenders similar effects to propofol on cortico-hippocampal interactions. 

Please also find the following some minor comments:

  • In lines 81-82, please justify the different time used to determine RORR.
  • In line 98, please define PSD.
  • In line 157, should it be “…comparison of PSD, PLV and MI at…”?
  • In line 208, please define the frequency range ofq-oscillation. 
  • In lines 287-288, was the animal anesthetized in a head-restraint apparatus?
  • Miscellaneous issues: Line 37, 58, 218 and lines 376-378.

Author Response

Dear Norah Tang,

Thank you for sending us the reviewers’ comments on our manuscript ijms-891696 entitled "Propofol affects cortico-hippocampal interactions via β3
subunit-containing GABAA receptors".

We thank the reviewers for providing valuable advice.  Here we present a revised version of our manuscript, which addresses the weaknesses pointed out by the reviewers. All changes in the manuscript are highlighted by using the “Track Changes” function.

We would like to answer the reviewer’s comments as follows:

Reviewer #2:

This study carried out by Kreuzer et al. investigated the potential molecular target of amnesia evoked by the general anesthetic propofol.  By recording local field potential (LFP) from the cortex and hippocampus in wild type and b3(N265M) knock-in mice, they observed that propofol differentially affects the power spectral density, phase locking value and mutual information in these two groups of mice.  The authors concluded that the frequency-specific perturbation of cortico-hippocampal interactions contributes, at least in part, to amnestic effect of propofol and that this effect is dependent on the GABAA receptor b3 subunit.  This is an interesting study.  The manuscript is well organized and presented. 

However, previous electrophysiology studies indicate that b2/b3 N265M mutation eliminates etomidate sensitivity and reduces propofol effects in a1b2/3g2 GABAA receptors.  Why was propofol instead of etomidate chosen in the current study?  It would be interesting to know if etomidate engenders similar effects to propofol on cortico-hippocampal interactions. 

Thank you for this suggestion. The reviewer is absolutely right. The b3(N265M) knock-in point mutation basically “works” for etomidate and propofol as well. While from a clinical point of view etomidate and propofol are rather similar, the molecular spectrum of propofol is slightly broader that that of etomidate: while propofol acts via GABAA receptors harboring any kind of b-subunit, etomidate favors b2- and b3- over b1-containing GABAA receptors [1-4]. This rather small difference in molecular targets nevertheless leads to pronounced differences between the electrophysiological „fingerprints“ of the two intravenous anesthetics [5].

In fact, we studied the effects of propofol AND etomidate in wild type and b3(N265M) mice using implanted cortical/hippocampal electrodes. However, the focus of the study with etomidate was slightly different from our current study and also was published some time ago [6]. The focus of the etomidate-study was more on the time period during and after awakening from etomidate anesthesia. The main findings were that i) in general the GABAA receptor b3(N265M) mutation largely attenuates the effects of etomidate on brain electrical activity, ii) during emergence from etomidate anesthesia the power density of oscillations in the range of 5 - 15 Hz transiently increased in the hippocampus of wild type mice, but not in the b3(N265M) mutant, and iii) the reduction of power in all frequency bands between 0.5 and 35 Hz by etomidate in hippocampus and cortex was less pronounced in GABAA receptor b3(N265M) point mutated mice.

Please also find the following some minor comments:

In lines 81-82, please justify the different time used to determine RORR.

First of all, animals were head fixed for electrophysiological recordings using implanted electrodes in the cortex and in the hippocampus. Therefore we had to detect basic behavioral parameters like e.g. return of righting reflex (RoRR) in separate, preceding experiments, please see line 80 et seqq. of our manuscript: „Therefore, time to RoRR after propofol injection was measured in separate experiments.”

As the GABAA receptor b3(N265M) point mutation renders this subtype of GABAA receptors more or less unresponsive to propofol the behavioral responses of knock-in mutant mice are quite different from the wild type. This has been originally shown in the paper by Jurd et al. [7]. The reported values in our current study lie approximately in the same range.

In line 98, please define PSD.

Thank you for this suggestion. We added the information and we also added a definition of all abbreviations used in the figure legends.

In line 157, should it be “…comparison of PSD, PLV and MI at…”?

Correct! We added “spectral power, PLV, and MI

In line 208, please define the frequency range ofq-oscillation. 

We added “relating to the theta range”.

In lines 287-288, was the animal anesthetized in a head-restraint apparatus?

Indeed, the reviewer is right, we apologize for this inaccuracy and added the information “in a head-restraint apparatus.” according to the reviewer’s suggestion in line 287-288.

Miscellaneous issues: Line 37, 58, 218 and lines 376-378.

Line 37: corrected; see comment to reviewer # 1.

Line 58: We corrected the numbering of the references

Line 218: corrected; see comment to reviewer # 1.

Lines 376 - 378: Thank you for this hint, we corrected this sentence.

Matthias Kreuzer and Berthold Drexler on behalf of the authors

Literature:

  1. Hill-Venning, C.; Belelli, D.; Peters, J. A.; Lambert, J. J., Subunit-dependent interaction of the general anaesthetic etomidate with the gamma-aminobutyric acid type A receptor. Br. J. Pharmacol 1997, 120, (5), 749-756.
  2. Tomlin, S. L.; Jenkins, A.; Lieb, W. R.; Franks, N. P., Stereoselective effects of etomidate optical isomers on gamma-aminobutyric acid type a receptors and animals. Anesthesiology 1998, 88, (3), 708-717.
  3. Belelli, D.; Muntoni, A. L.; Merrywest, S. D.; Gentet, L. J.; Casula, A.; Callachan, H.; Madau, P.; Gemmell, D. K.; Hamilton, N. M.; Lambert, J. J.; Sillar, K. T.; Peters, J. A., The in vitro and in vivo enantioselectivity of etomidate implicates the GABAA receptor in general anaesthesia. Neuropharmacology 2003, 45, (1), 57-71.
  4. Sanna, E.; Murgia, A.; Casula, A.; Biggio, G., Differential subunit dependence of the actions of the general anesthetics alphaxalone and etomidate at gamma-aminobutyric acid type A receptors expressed in Xenopus laevis oocytes. Mol. Pharmacol 1997, 51, (3), 484-490.
  5. Drexler, B.; Jurd, R.; Rudolph, U.; Antkowiak, B., Distinct actions of etomidate and propofol at beta3-containing gamma-aminobutyric acid type A receptors. Neuropharmacology 2009, 57, (4), 446-455.
  6. Butovas, S.; Rudolph, U.; Jurd, R.; Schwarz, C.; Antkowiak, B., Activity patterns in the prefrontal cortex and hippocampus during and after awakening from etomidate anesthesia. Anesthesiology 2010, 113, (1), 48-57.
  7. Jurd, R.; Arras, M.; Lambert, S.; Drexler, B.; Siegwart, R.; Crestani, F.; Zaugg, M.; Vogt, K. E.; Ledermann, B.; Antkowiak, B.; Rudolph, U., General anesthetic actions in vivo strongly attenuated by a point mutation in the GABA(A) receptor beta 3 subunit. FASEB Journal 2003, 17, (14), 250-252.